

# 1 Impact of lower atmospheric scattering on ground-based optical
# 2 thermospheric wind observations with spatially uneven airglow

Xiaolong Wei[1,2,4], Guoying Jiang[1,3,4,*], Yajun Zhu[1,3,4,*], Jiyao Xu[1,4], Weijun Liu[1,4], Tiancai Wang[1,2,4],
Guangyi Zhu[1,4], Wei Yuan[1,4]
[1]State Key Laboratory of Solar Activity and Space Weather, National Space Science Center, Chinese Academy of Sciences,
Beijing, 100190, China
[2]College of Earth and Planetary Sciences, University of Chinese Academy of Sciences, Beijing, 100049, China.
[3]School of Astronomy and Space Science, University of Chinese Academy of Sciences, Beijing, 100049, China.
[4]Hainan National Field Science Observation and Research Observatory for Space Weather, Hainan, 571734, China
*Correspondence to: Guoying Jiang (gyjiang@spaceweather.ac.cn), Yajun Zhu (y.zhu@swl.ac.cn)
**Abstract.** Scattered airglow emissions in the lower atmosphere can bias ground-based interferometer observations of
thermospheric winds, particularly when airglow brightness becomes spatially uneven due to auroras. During two geomagnetic
storms with visible auroras on May 10[th] and Oct. 10[th], 2024, the Doppler Asymmetric Spatial Heterodyne (DASH) and Fabry-
Perot (FP) interferometers concurrently detected atypical winds at Siziwang (SIZW, 41.83° N, 111.93° E), suspected to be
caused by scattering. These atypical winds, characterized by horizontal differences exceeding 400 m·s⁻¹ between opposite
cardinal directions (N-S or E-W) and downwelling exceeding 100 m·s⁻¹, showed a strong temporal association with airglow
brightness. By modelling the transmission of scattered airglow emissions, we calculated post-scattering wind speeds as the
initial wind speeds weighted by both scattered and direct intensities. With fixed initial speeds, the simulation reproduced the
temporal characteristics of the atypical winds, demonstrating that scattering causes these intense horizontal differences and
downwelling. The simulation also shows that the scattering-induced biases have directional inhomogeneity with characteristics
linked to the location and background line-of-sight speed of the brighter airglow region. The commonly used horizontal
average wind may experience numerical deviations due to directional inhomogeneity. The accuracy of the simulation is limited
by the accuracy of airglow observations and atmospheric optical depth.

## 24 1 Introduction

Optical interferometers are widely utilized to observe thermospheric neutral wind (Burnside et al., 1981; Burnside and Tepley,
1989; Killeen et al., 1995; Emmert et al., 2001; Fejer et al., 2002; Emmert et al., 2006; Wu et al., 2014). Thermospheric wind
can be derived from measuring the Doppler shift of OI red-line airglow emission at 630.0 nm. This emission, primarily from
the collisional deactivation of O($^1$D) generated by O$_2^+$ dissociative recombination, peaks near 250 km altitude. The height-
integrated thermospheric wind around the peak altitude can be obtained (Biondi and Feibelman, 1968; Hernandez and Roble,
1976; Burnside et al., 1981; Biondi et al., 1995; Nakajima et al., 1995). For scanning interferometers, three-dimensional wind
vectors can be derived by observing the zenith and four cardinal directions at a specific elevation angle. The scanning range



covers a circular area about 500 km in diameter at airglow altitude. Given thermospheric wind uniformity at this scale,
horizontal winds observed in two opposite cardinal directions (N-S or E-W) are typically similar. Averaging opposite cardinal
directions improves accuracy, mitigates cloud effects, and is typically used to represent local meridional or zonal winds even
during geomagnetic storms (Friedman and Herrero, 1982; Fejer et al., 2002; Sakanoi et al., 2002; Dhadly et al., 2017; Huang
et al., 2018; Xu et al., 2019; Li et al., 2023; Wang et al., 2025).
However, horizontal winds in opposite cardinal directions occasionally show significant separation exceeding 100 m·s$^{-1}$, with
strong vertical winds generally deviating from typical thermospheric wind uniformity. These observations often occur near
auroras, unaffected by clouds or moonlight, and have acceptable standard errors. They mainly occur in polar regions
(Crickmore et al., 1991; Price et al., 1995; Smith and Hernandez, 1995; Innis et al., 1999; Ishii et al., 2001; Guo and Mcewen,
2003; Anderson et al., 2012), but have also been seen at mid-latitudes during major magnetic storms (Hernandez and Roble,
1976; Makela et al., 2014).
Atmospheric scattering of airglow emissions introduces errors to ground-based interferometers, potentially accounting for
these atypical wind observations. Initially, it was thought to impact airglow peak height measurements by photometers
(Ashburn, 1954). Subsequent studies by Abreu et al. (1983) explored its impact on thermospheric wind speed measurements
using a Fabry-Perot interferometer. Harding et al. (2017a; 2017b) later systematically modelled and estimated these effects,
revealing that scattering was responsible for the anomalous vertical winds observed at mid-latitudes during geomagnetic storms
by Makela et al. (2014). Light from brighter airglow regions scatters omnidirectionally in the lower atmosphere, primarily the
troposphere and stratosphere, and is detectable outside its original direction. The additional Doppler shift of this scattered light
can bias the retrieval of line-of-sight (LOS) speeds as well as the converted horizontal and vertical winds. Scattering-induced
biases are more pronounced during spatially uneven airglow brightness, such as during auroras (Harding et al., 2017a). Uneven
airglow brightness refers specifically to inhomogeneous red-line emissions. Airglow and auroral emissions have similar
wavelengths and peak heights, though differing in mechanism, making them hard to distinguish in ground-based observations.
Harding et al. (2017b) also investigated the impact of atmospheric scattering on interferometer wind and temperature
measurements during quiet periods and attempted to correct for the associated errors. Additionally, spectral contamination
from precipitating energetic ions can also bias interferometers (Makela et al., 2014). They suggested that the enhanced
downwelling at mid-latitudes during storms might result from the contamination of the spectral profile by fast O atoms
associated with the influx of low-energy O$^+$ ions.
From a dynamical perspective, wind differences in opposite cardinal directions are considered horizontal divergence, which
are often associated with changes in vertical winds. Near the aurora arc, these atypical winds are mainly caused by ion drag,
Joule heating, and energy particle precipitation (Hays et al., 1984; Rees et al., 1984; Conde and Smith, 1995; Conde et al.,
2001; Anderson et al., 2012). Generally, excessive horizontal divergence and vertical wind appear alongside rapidly changing
auroras and exhibit a matching spatial relationship that upward (downward) winds accompanied by divergences (convergences)





are often detected when aurora exists equatorward (poleward) of the observatory (Ishii et al., 2001; Guo and Mcewen, 2003).
The combination of vertical wind and horizontal divergence is related to gravity waves excited by the above processes in polar
regions, presenting a wave-like structure and phase delay between vertical and horizontal wind components. (Price et al., 1995;
Smith and Hernandez, 1995; Ishii et al., 1999; Ishii et al., 2001; Shinagawa and Oyama, 2006). At mid-latitudes, which are
not primary regions for magnetospheric energy injection, atypical winds are instead related to the propagation of gravity waves
from polar regions. (Hernandez and Roble, 1976).
During two geomagnetic storms with visible auroras, we observed similar atypical winds in ground-based interferometers at
Siziwang (SIZW, 41.83° N, 111.93° E), China. These winds showed intense differences over 400 m·s⁻¹ in two opposite cardinal
directions for both meridional and zonal components, along with downward wind exceeding 100 m·s⁻¹. The observations were
unaffected by moonlight or clouds, and the interferometer retrieval errors were acceptable. These atypical winds at SIZW only
occurred with auroras and significantly deviated from the regional climatological norms over the China region (Jiang et al.,
2018; Yang et al., 2020). This raises the question of whether the atypical winds are driven by dynamical processes or by
scattering-induced biases of interferometers, which causes the thermospheric wind to deviate from horizontal uniformity.
Unfortunately, both factors can manifest as increased differences between opposite cardinal directions, complicating the
distinction between them (Harding et al., 2017a). However, the simultaneous variations in vertical winds, horizontal
differences, and red-line brightness show no phase lag, thus not providing evidence for the energy conversion process (Ishii et
al., 1999). Instead, these variations resemble a systematic error, as they all involve negative LOS speeds. This suggests that
scattering impact may be more significant than dynamic mechanisms in these cases.
Auroras are rare at mid-latitudes in East Asia due to low magnetic latitude. The magnetic latitude of SIZW is just 37.7° N. But
increased solar activity has led to more frequent sightings of mid-latitude red auroras in Japan and China (Kataoka et al., 2024a;
Kataoka et al., 2024b; Ma et al., 2024). Previously overlooked, the scattering-induced biases associated with auroras need
further analysis to determine the accuracy and suitability of storm-time observations for dynamical analysis. While prior
studies have examined vertical wind biases of Fabry-Perot interferometers under auroral conditions (Abreu et al., 1983;
Harding et al., 2017a; Harding et al., 2017b), we will incorporate Doppler Asymmetric Spatial Heterodyne (DASH)
interferometer data to compare scattering impact across different interferometer types. Additionally, we will investigate the
causes and patterns of horizontal differences and assess the reliability of individual directions and horizontal average winds
under the influence of scattering. In the following text, a scattering radiative transfer model is used to simulate interferometer
observations in two cases with visible aurora. The presence and patterns of scattering-induced biases are analyzed by
comparing simulations with observations.



## 2 Instruments and model

This study was conducted at the Siziwang station (SIZW; 41.83° N, 111.93° E, and 37.7° N MLat) of the Chinese Meridian Project Phase II (Wang et al., 2024), utilizing a Dual-Channel All-sky Airglow Imager (DCAI), a Dual-Channel Optical Interferometer (DCOI), and a Fabry-Perot Interferometer (FPI). DCOI derives neutral winds by observing atomic oxygen green-line (557.7 nm, around 96 km) and red-line (630.0 nm, around 250 km). DCAI observes hydroxyl (around 87 km) and atomic oxygen red-line nightglow, respectively. FPI only works at the red-line. Our focus is on the red-line channel. Using DCAI images as one of the inputs, wind biases from optical interferometers can be simulated by a scattering radiative transfer model (scattering model for short). Instruments and the model are described in the following subsections.

### 2.1 Dual-Channel All-Sky Airglow Imager

Dual-Channel All-Sky Airglow Imager (DCAI) comprises a fisheye lens with an approximate 170 degree field of view, a 2 nm narrow-band filter, and a 1024×1024 pixel, 16 bit cooled CCD. DCAI exposure time of the red-line is 2 minutes. The obtained airglow images are first calibrated to the local spherical coordinate system, then sequentially corrected for stray light, Van Rhijn effect, and atmospheric extinction, and finally projected onto the 250 km airglow plane. Due to DCAI not calibrating the Rayleigh unit (Shiokawa et al., 2000), observed brightness is only normalized to the full-well value. And because of fish-eye lens distortion and the lack of Rayleigh unit calibration, the edge brightness of the view is inaccurate. Thus, observations are restricted within a 70° zenith angle. For larger zenith angles, the brightness is obtained by radial zero-order extrapolation in airglow projection. Detailed image processing procedures are in Appendix B.

### 2.2 Dual-channel optical interferometer

Dual-channel optical interferometer (DCOI) is a scanning interferometer using Doppler Asymmetric Spatial Heterodyne (DASH) technology. DASH exhibits a wider field of view, better thermal stability, simplified mechanisms, and lower tolerance requirements than other interferometer structures (Englert et al., 2007; Englert et al., 2010; Harlander et al., 2017; Wei et al., 2020). DCOI consists of a 630 nm narrow-band filter (2 nm bandwidth), a 9 degree field-of-view lens (f/6), a DASH interferometer with a 25 mm aperture, a Neon lamp for calibration, and a 2048×2048 pixel CCD (13.5 μm per pixel) (Wei et al., 2020; Zhu et al., 2023; Liu et al., 2025). Its thermal stability is maintained within 0.1 K. DCOI measures three-dimensional wind speeds by scanning five directions (zenith and four cardinal directions at 45° zenith angle). Each direction is exposed for 5 minutes, completing a cycle roughly every 25 minutes. DCOI adopts an observation with the smallest error after evening as the reference zero wind speed. The slant LOS speeds are subtracted by the time-regressed projection of vertical speed and then converted to horizontal using the sine of zenith angles. It is worth noting that during auroral events, vertical winds with absolute values exceeding 50 m·s⁻¹ are excluded from the regression, as they contain scattering effects that could introduce additional biases to other directions. DCOI provides two series of meridional wind, two series of zonal wind, and one series of vertical wind.



### 2.3 Fabry-Perot Interferometer

Fabry-Perot Interferometer (FPI), as a mature solution, conducts comparative observations with DCOI. It features a 630 nm narrow-band filter (2 nm bandwidth), a 2.54 degree field-of-view lens (f/6), a 50 mm aperture etalon with a 7 mm gap, a frequency-stabilized laser for calibration, and a 1024×1024 pixel CCD (13 μm per pixel). FPI uses the same integration time and scanning method as DCOI to obtain horizontal and vertical winds for each cardinal direction and zenith. Details and historical results of FPI are in these references (Yuan et al., 2010; Wu et al., 2014; Yu et al., 2014; Huang et al., 2018; Jiang et al., 2018).

### 2.4 Scattering radiative transfer model

The model for estimating scattering impact is based on the scattering radiative transfer model and numerical solution by Harding (2017a). It assumes airglow emission undergoes elastic scattering, preserving its wavelength and initial Doppler shift. By specifying airglow brightness distribution, original Doppler shift, lower atmosphere scattering characteristics, and a simplified geometric relationship, the radiation transfer equations (see Appendix A) can be solved to compute the distribution of multiple scattered light and its associated Doppler shift. This enables the wind simulation with atmospheric scattering. A schematic diagram (Fig. 1) illustrates the basic mechanism. To enhance applicability, we have refined several aspects: (1) The upper boundary of the lower atmosphere is set at 40 km to improve the accuracy of the effective extinction path in the initial source function. (2) The LOS speed is used directly instead of the Doppler shift, assuming a constant background temperature. (3) After binning different LOS speeds and computing the corresponding scattered light intensity, contaminated LOS speeds are calculated via weighted average, simplifying the wind simulation. We directly use LOS speed instead of Doppler shift, primarily neglecting the interference fringe recognition errors caused by spectral broadening due to temperature variations. During auroral events, FPI observations show similar neutral temperatures in all directions, with the northward direction occasionally being about 300 K higher (not shown here). Overall, the temperature at mid-latitudes is uniform at the 500 km spatial scale, and the variations caused by spectral line broadening can be neglected. The detailed model description is provided in Appendix A.

Additionally, the scattering characteristics of the lower atmosphere in our model, including the scattering phase function and optical depth, were derived from Aerosol Robotic Network (AERONET) observations (Holben et al., 2001). We utilized data from the Baotou site (40.9° N, 109.6° E), which is the nearest available site to SIZW, located approximately 180 km away. The total optical depth, accounting for both aerosol and molecular scattering, was calculated using monthly averages and was found to be 0.43 in May and 0.2 in October. The scattering phase function was determined based on AERONET data following the previous method (Harding et al., 2017a). Further details regarding the scattering characteristics are described in Appendix B.



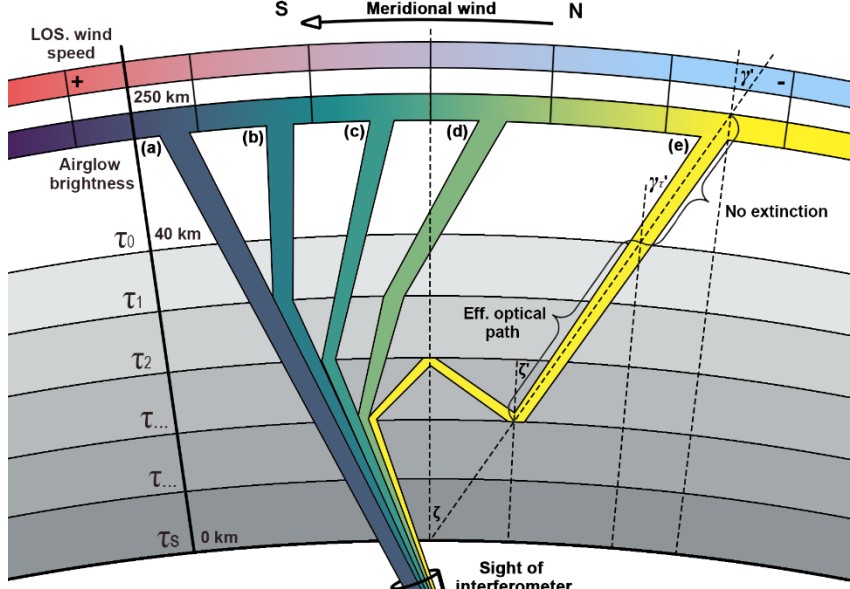

**Figure 1: The schematic diagram of the scattering radiative transfer model**

The grey shading represents the lower atmospheric layer, with darker hues indicating greater optical depth. The yellow-green fillers represent the relative brightness from the red-line airglow layer. Yellow indicates higher light intensity. The blue-red fillers, which correspond to the relative brightness, represent the Doppler shift type (blue-shift or red-shift) of LOS wind speeds. (a) to (e) represent airglow emissions travelling along different paths, carrying Doppler shifts from outside the line of sight into the interferometer, thereby causing biases in the observations. The model estimates the biases by simulating the distribution of airglow emissions after scattering.

## 3 Results

Two storms with visible auroras on May 10[th] and Oct. 10[th], 2024, respectively, are used to study the scattering impact. The storm from May 10[th] to 11[th] is characterized by its significant magnitude and prolonged duration. Multiple works report this event (Guo et al., 2024; Hajra et al., 2024; Themens et al., 2024), with particular focus on the variations of thermospheric winds (Wang et al., 2025; Zhang et al., 2025) and auroras (Gonzalez-Esparza et al., 2024; Kataoka et al., 2024b; Mikhalev, 2024; Nanjo and Shiokawa, 2024) at mid-latitudes. The storm commenced around 17:00 UT on May 10[th] and the main phase persisted until 02:00 UT on May 11[th]. After that, the local night of May 11[th] in the China region sank into a continuous recovery phase. Another storm from Oct. 10[th] to 11[th] is weaker than May's (Ranjan and Pallamraju, 2025; Singh et al., 2025), with the main phase from 18:00 UT on Oct. 10[th] to 02:00 UT on Oct. 11[th]. During the two geomagnetic storms with visible auroras, both the DCOI and FPI at SIZW observed atypical winds, characterized by intense horizontal wind differences and downward vertical winds.



## 3.1 Storm-time wind speed statistics

It is necessary to ascertain whether atypical winds originate from atmospheric scattering with spatially uneven airglow brightness or dynamic processes during storms. To investigate the impact of visible auroras on atypical winds during storms, we made the most of the available observations, tracking DCOI's storm-time observations for nearly a year and FPI's for almost five months. We employed the planetary magnetic index Kp exceeding 3 to identify geomagnetic storms (Yang et al., 2020). Besides, to rule out moonlight and cloud effects, we only used clear sky conditions, which means: (1) excluding cases where the angle between the moon and the line of sight is less than 30 degrees, and (2) excluding cases where large-area thick cloud coverage is visible in DCAI. Additionally, data with standard errors greater than 50 m·s⁻¹ were also excluded. A few aurora events, including Nov. 5th, Dec. 1st, 2023, and Aug. 12th, 2024, that did not meet this criterion were excluded.

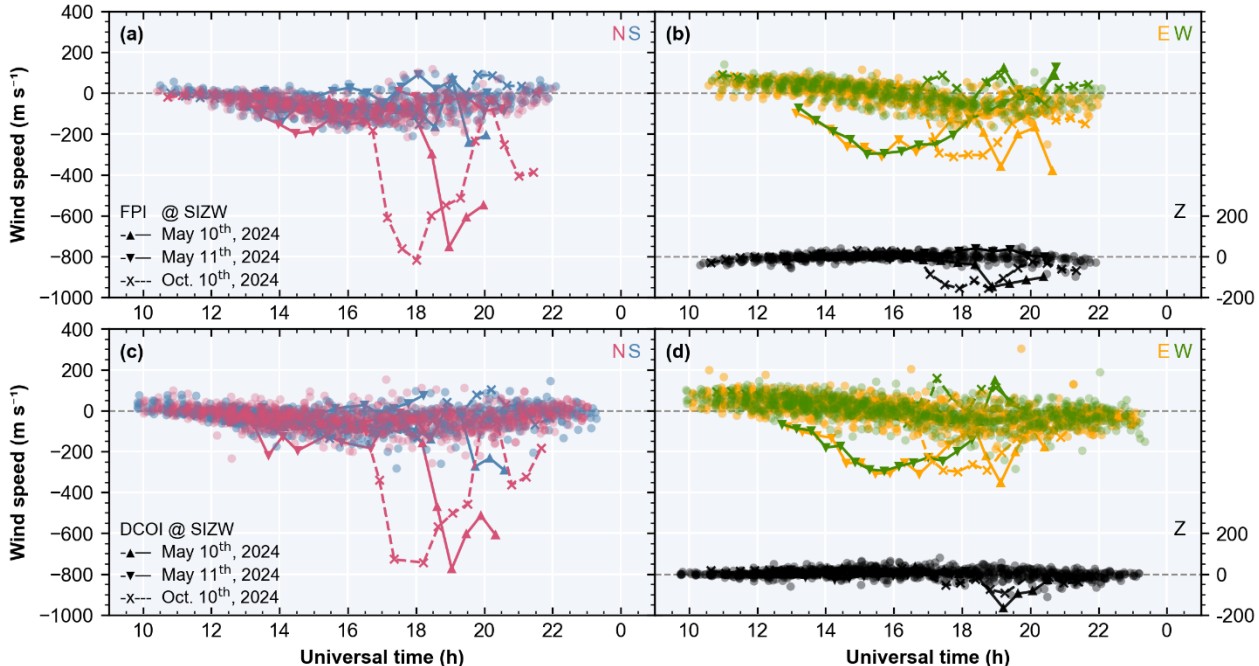

**Figure 2: Storm-time (Kp>3) thermospheric wind speed statistics at SIZW**

Figure 2a shows meridional winds observed along two opposite directions (N-S) by FPI, with north-looking in red and south-looking in blue. Figure 2b shows zonal and vertical wind, with east-looking in yellow, west-looking in green, and zenith-ward in black. The northward, eastward, and upward speeds are positive in coordinates. Observations without aurora are shown as points, while those with visible auroras are shown as lines. Figures 2c and 2d are similar but show DCOI data.

Figure 2 shows thermospheric wind statistics during geomagnetic storms (Kp>3) at SIZW. The first two panels display FPI data from May to Oct. 2024, while the rest display DCOI data from Nov. 2023 to Oct. 2024. FPI began operation on May 8th, 2024, with about half a year less data than DCOI. Observations with no aurora in the field of view are marked as points, while



the two cases with visible auroras are shown as lines. The five observation directions of the interferometer are marked by
different colors. During typical storms, horizontal winds consistently increase to around 150 m·s⁻¹ both equatorward and
westward with no significant downward wind. However, under visible auroras, both DCOI and FPI have detected large wind
speeds, such as a southward wind of about 600 m·s⁻¹ and downwelling exceeding 100 m·s⁻¹. The two series of winds observed
along opposite cardinal directions (N-S or E-W) exhibit overt differences, with values exceeding 400 m·s⁻¹ and contrary
directions. This is markedly different from the wind patterns observed during non-aurora storms, where opposite-direction
winds do not show significant divergence. Comparing the results of DCOI and FPI, the observations are largely consistent
both with and without auroras. The atypical winds observed simultaneously by two interferometers with different principles
suggest a systematic error from outside the instruments. Besides, these simultaneous changes appear in five observation
directions, all characterized by enhanced negative LOS speeds, indicating likely LOS speed contamination. These factors point
more towards scattering impact rather than dynamical processes as the cause. Next, the relationship between scattered light
and atypical winds will be investigated through simulation.

**3.2 Comparison of observations and simulations**

Figure 3 shows the red-line airglow brightness from DCAI (Fig. 3a, 3e), the observed winds from DCOI and FPI (solid lines
with different markers in Fig. 3b-3d, 3f-3h) and the simulated winds from the scattering model (dotted lines in Fig. 3b-3d, 3f-
3h) during the two nights of May 10th and 11th, 2024, at SIZW, in which the different colors denote distinct directions. The
grey lines in the horizontal wind plots represent the average values between opposite cardinal directions. The multi-directional
brightness series from DCAI are extracted at 45° zenith angle, consistent with the scanning zenith angle of interferometers.
Time intervals with visible auroras are highlighted in red, showing much higher brightness in northward directions than others.
Figure 4 supplements the auroral distribution compared to Fig. 3a and 3e. Images from DCAI are projected onto the airglow
layer at 250 km. The red circle encloses the actual observations with zenith angles less than 70°, while the values outside are
extrapolated. The red dots represent the interferometer's pierce points on the airglow layer at 45° zenith angle.





**Figure 3: Observations of aurora and wind speeds, and the scattering model simulation on the nights of May 10th and 11th, 2024, at SIZW**

Figure 3a shows the brightness of 8 cardinal directions, all at 45° zenith angle, along with the zenith-ward, extracted from DCAI. The color coding is as follows: red for northern directions, green for east and west, blue for southern directions, black for the zenith, and yellow for the average brightness excluding the three northern directions. Figure 3b shows the meridional wind, with north-looking in red and south-looking in blue, and the average of the two directions in grey. DCOI observations are shown as solid lines with circular dots, FPI as solid lines with rhombus dots, and simulations as dotted lines. Figures 3c, 3d are similar to Fig. 3b, but for zonal and vertical wind, with east-looking in yellow, west-looking in green, and zenith-ward in black. For a more concise figure, if the standard error exceeds 100 m·s⁻¹, the point will be filled with black instead of error bar. Figures 3a-3d show data from May 10th, and Fig. 3e-3h from May 11th. The time intervals with visible auroras are marked in red.





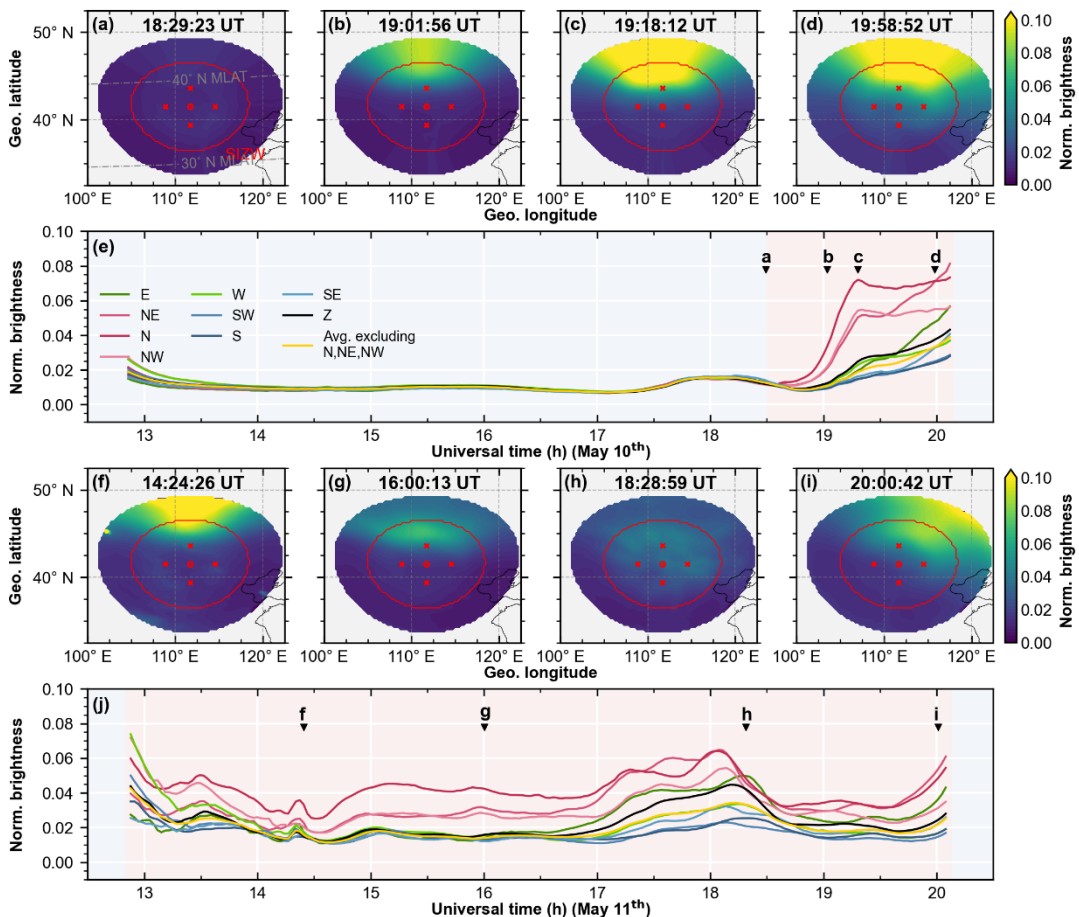

**Figure 4: Auroral distribution observed by DCAI on the nights of May 10th and 11th, 2024, at SIZW**

Images from DCAI (Fig. 4a-4d, 4f-4i) have been projected onto the airglow layer at 250 km. The red circle encloses actual observations with zenith angles < 70°, while values outside are extrapolated using zero-order extrapolation. The red dots represent the interferometer's pierce points on the airglow layer at 45° zenith angle. Figures 4e, 4j are similar to Fig. 3a, 3e, with the corresponding images time labelled. The coastlines in projected DCAI images are made with Natural Earth.

After 18:30 UT on May 10th, as the aurora intensified, both DCOI and FPI detected simultaneous changes in meridional, zonal, and vertical winds. The north-looking red-line brightness at 45° zenith angle exceeded three times that of other directions. The meridional and zonal wind differences between opposite cardinal directions (N-S or E-W) increased. And the winds detected in opposite directions reversed. The maximum meridional difference was close to 800 m·s⁻¹, while that in zonal exceeded 500 m·s⁻¹. The downward wind was enhanced by over 100 m·s⁻¹. These four aforementioned variables, that red-line brightness, the meridional and zonal differences, and downwelling, increased almost simultaneously, peaked at 19:05 UT, and then decayed.





Moreover, the average meridional wind, derived from averaging opposite cardinal directions, continuously enhanced
equatorward to around 400 m/s, while the average zonal wind enhanced westward to around 100 m/s. Unlike the single-
direction results that peaked at 19:05 UT, the average wind varied steadily, consistent with storm-time circulation. Compared
to the average wind, the separated horizontal and enhanced vertical winds are atypical. Even with travelling atmospheric
disturbances (TADs) superimposed on storm-time circulation, phase lags between horizontal and vertical components would
be expected (Hernandez and Roble, 1976; Ishii et al., 1999), but none were observed. Thus, the atypical winds do not appear
to be the result of a dynamical process. During the recovery phase on May 11th, the aurora was present throughout the night
but much weaker than on May 10th, as seen in Fig. 4. Both DCOI and FPI showed westward and equatorward horizontal winds
with no significant downward wind. There was a meridional difference of about 100 to 300 m·s⁻¹ persisted throughout the
night, with no zonal difference.
Subsequently, we used the scattering model to explore the relationship between red-line brightness variations and atypical
winds through atmospheric scattering. On May 10th, a fixed wind vector of 100 m·s⁻¹ westward and 400 m·s⁻¹ southward was
set as the input. This assumed wind was kept constant over time and spatially uniform, with no vertical components. On May
11th, a fixed wind vector of 200 m·s⁻¹ westward and 100 m·s⁻¹ southward was used, again with no vertical component. These
values are chosen based on average observed wind speeds to approximate storm-time circulation. Although the specific values
may deviate, the main wind directions remain consistent. The storm-time enhancement of vertical winds may be caused by
scattering rather than representing real winds, so we set it to zero. It is worth noting that we neglect the variation of background
wind in the model inputs, due to uncertainty regarding whether the observed wind variations are biased. Moreover, using fixed
wind speeds allows us to highlight the impact of red-line brightness variations and determine the presence of scattering effects.
As the dotted lines in Fig. 3b-3e, the simulations with scattering impact generally match the observations on May 10th.
Simulated horizontal wind differences and downwelling increase initially with the aurora, peak around 19:15 UT, and then
decay. The simulated wind speed variations lag the observations by about 10 minutes. The lag may be due to the relatively
rough 25 minute scanning cycle or DCAI underestimating airglow brightness at the field of view's edge, leading to inaccurate
capture of scattering enhancement start time. Numerically, the simulated zonal and vertical winds match observations more
closely than the meridional wind. The simulated meridional difference is smaller than the observed difference, and the north-
looking simulation remains closer to the default value, unlike the equatorward-biased observation. The preset fixed wind may
influence the meridional simulation, as it does not follow the equatorward enhancement of the meridional wind. Other factors
beyond scattering impact might also have an impact, such as spectral pollution caused by auroras (Makela et al., 2014), to be
discussed later. Additionally, the model simulates a similar intense downward wind as observed under the preset zero vertical
wind. This indicates that the vertical wind is significantly affected by scattering. This is why the intense vertical wind is not
subtracted when converting interferometer LOS speed to horizontal wind, to prevent error propagation. For May 11th, due to
the weak but continuous aurora, the simulation shows weak horizontal differences and slight downward winds throughout the



night. Compared with the observations, the simulation shows smaller meridional differences. It also indicates zonal differences
and downward winds, which are not evident in the observations. The poorer simulation on May 11th may be due to
misalignment between dominant horizontal winds and airglow brightness gradients, which will be discussed later. Additionally,
there may be issues with the zero wind calibration. When auroras are present throughout the night, the vertical wind, which
includes scattering biases, may have been used to calibrate zero wind speed. It likely masks the scattering impact in the
observations and explains the discrepancies in the simulation.

**Figure 5: Observations of aurora and wind speeds, and the scattering model simulation on the nights of Oct. 10th and 11th, 2024, at SIZW**

Figure 5 is analogous to Fig. 3, but for Oct. 10th and 11th.



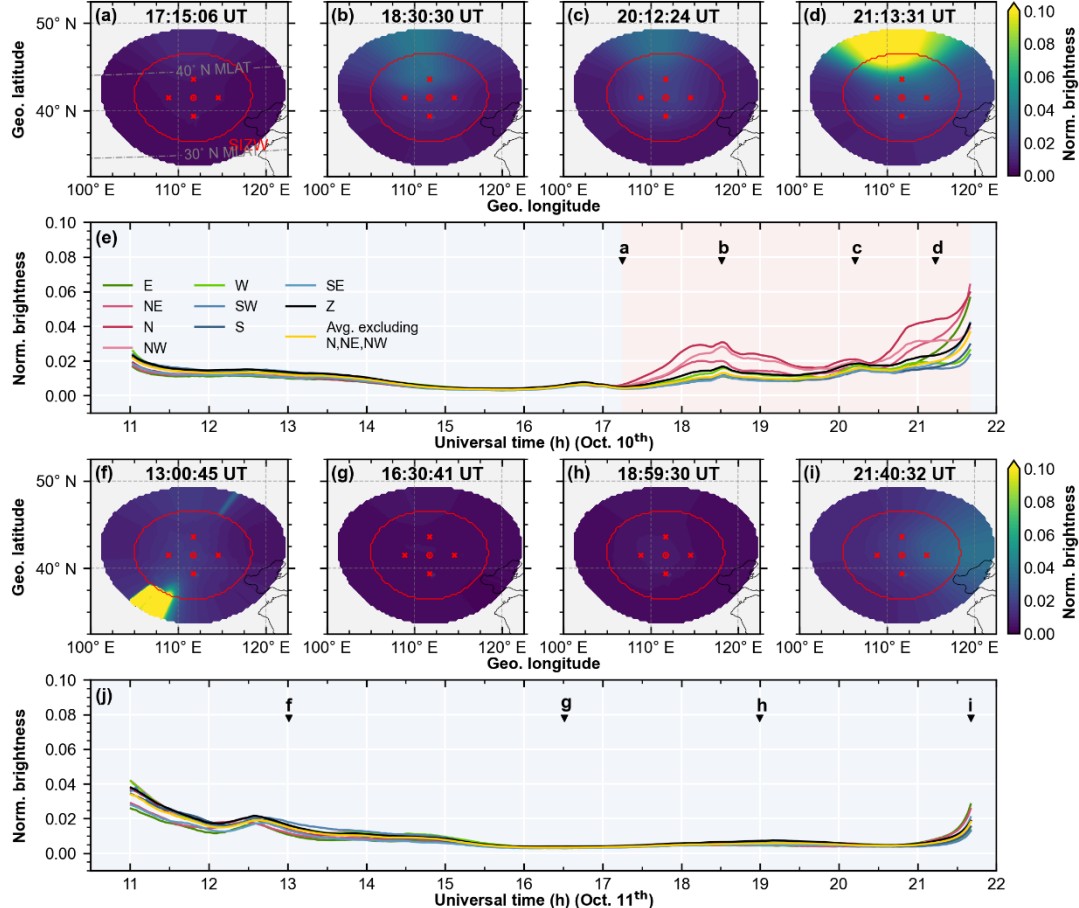

**Figure 6: Auroral distribution observed by DCAI on the nights of Oct. 10th and 11th, 2024, at SIZW**

Figure 6 is analogous to Fig. 4, but for Oct. 10th and 11th.

Figures 5 and 6 show another case from Oct. 10th to 11th, analogous to Figures 3 and 4. On Oct. 10th, the aurora appeared at 17:15 UT, expanded southward and increased in brightness, peaking first at 18:30 UT before decaying and then increasing again from 20:30 UT until sunrise. The second peak was brighter than the first (Fig. 6). Similar to the storm in May, once aurora appeared, both DCOI and FPI observed atypical winds, with synchronous meridional and zonal differences and downward enhancements in vertical wind. These atypical winds also exhibited two peaks, around 18:30 UT and 21:00 UT. The horizontal winds observed in opposite cardinal directions were basically in opposition. During visible aurora periods, DCOI and FPI showed a 50 to 100 m·s$^{-1}$ difference in vertical wind but consistent variation trends. Moreover, the average horizontal wind between opposite cardinal directions was dominantly equatorward and westward, which also had two peaks. On Oct. 11th, the storm had passed, and no visible aurora was present. The increase in brightness around 13:00 UT was due to moonset in the southwest. There were no significant horizontal differences or downward winds, consistent with geomagnetic quiet conditions.



We set a fixed 100 m·s⁻¹ westward with 400 m·s⁻¹ southward wind vector for Oct. 10$^{th}$, and 100 m·s⁻¹ westward with 200 m·s⁻¹
southward wind vector for Oct. 11$^{th}$ in the model, respectively, with no vertical component. The simulation on Oct. 10$^{th}$ exhibits
two peaks. The second peak in the simulation is consistent with the observations better, while the first peak, although capturing
the trend, is significantly underestimated in magnitude. This discrepancy in the simulation may relate to optical depth, aurora
brightness, and background wind changes. The optical depth in Oct. is about half that of May, and simulations underestimate
observed values. As in previous studies (Harding et al., 2017b), optical depth can affect the scattering model response. On Oct.
10$^{th}$, the first aurora brightening is weaker than the second. When red-line brightness differences are small, the model response
tends to be lower. The impact of optical depth and red-line brightness on the model will be discussed later. Additionally,
noticeable fluctuations in the average meridional wind on this day may also contribute to the deviation in the model with fixed
initial wind. The north-looking wind speed varied dramatically between 19:00 UT and 21:00 UT along with the aurora, which
may also be related to spectral contamination beyond scattering impact. This spectral contamination arises from fast O atoms
generated by low-energy O⁺ ion precipitation in the auroral region, which occurs at higher altitudes. This issue introduces an
additional spectral shift that compromises wind retrieval (Makela et al., 2014). This exceeds the simulation range of the model,
thereby causing the discrepancy.

## 4 Discussions

In this study, we modelled scattered airglow transmission in the lower atmosphere. Post-scattering wind speeds were calculated
based on initial wind speeds weighted by both scattered and direct intensities. The model basically captured the temporal
variations of horizontal wind differences and downward enhancements associated with varying auroral brightness, suggesting
the contribution of scattering mechanisms to atypical winds. However, the simulation of scattering has certain limitations and
characteristics: (1) The differences between simulation and observation vary across different directions. (2) The simulated
values sometimes exhibit significant numerical deviations from observations. Could this be related to model errors? (3) When
using average wind speeds and individual direction wind speeds with scattering impact, what additional considerations are
needed? Next, we will discuss these three issues in detail, focusing on the working principle of the model, the errors involved,
and the insights that the simulation can provide regarding the impact of scattering.

### 4.1 Core working principle of the scattering model

As shown in Fig. 3 and Fig. 5, observed winds respond differently to scattering across directions, especially on May 10$^{th}$ and
Oct. 10$^{th}$ with stronger auroras. Although the scattering model has numerical errors, the simulations also show directional
differences in scattering-induced biases. Both observations and simulations indicate that the meridional wind responds the
most, followed by the zonal wind, while the vertical wind responds the least. Since vertical and horizontal wind speeds are
derived from the projection of LOS wind speeds, this essentially reflects the non-uniform response of LOS speeds to scattering.
This directional inhomogeneity of scattering impact aligns with previous studies. Harding et al. (2017a) simulated scattering




effects under auroral conditions, using northward observations as the initial winds. They noted that this direction experiences
minimal scattering contamination due to facing the brighter region. Abreu et al. (1983) used a meridional one-dimensional
model, finding that LOS wind speeds near intense airglow brightness gradients and with weaker airglow intensity are more
susceptible to contamination. They also showed that scattering-induced biases are minimal in the vertical direction because
the light path through the atmosphere is shortest, reducing scattering opportunities. In this study, we further explore the
scattering impact as a function of azimuth angle, revealing the formation of horizontal differences.

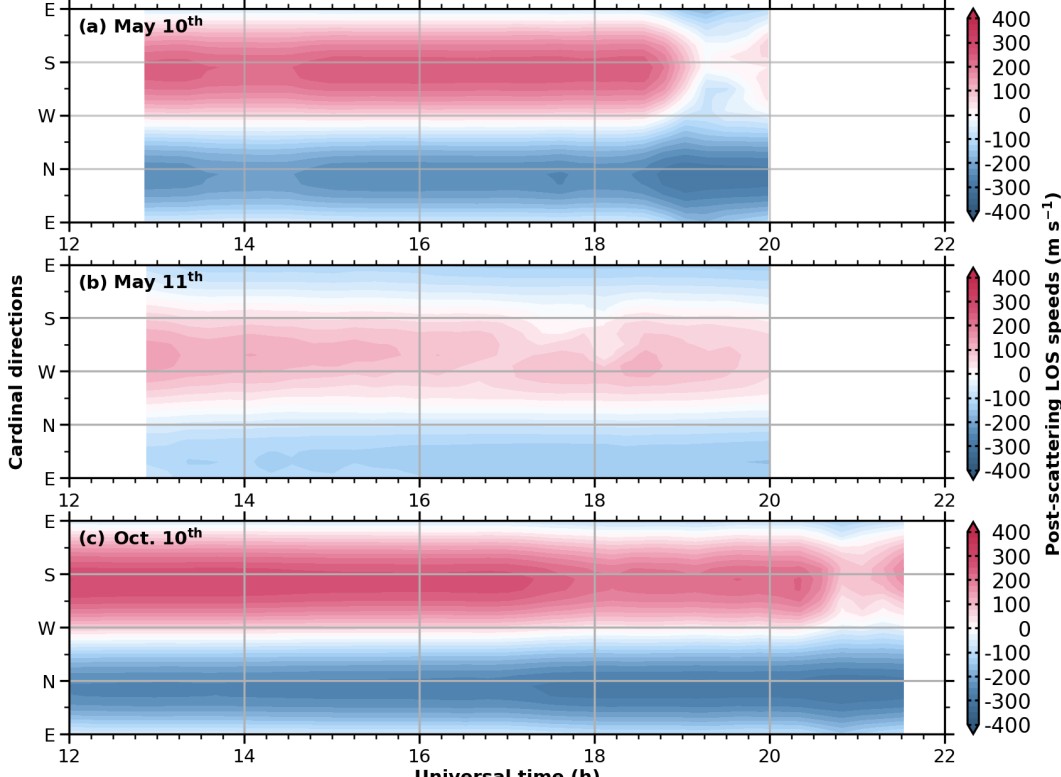


**Figure 7: Post-scattering LOS speeds at 45° zenith angle**
The figure shows the post-scattering line-of-sight speeds at 45° zenith angle for various directions over time, with panels for May 10th,
May 11th, and October 10th, respectively.
Figure 7 shows post-scattering LOS speeds at 45° zenith angle in the simulations on three aurora nights, with the vertical axis
indicating cardinal directions derived from azimuth angles. Concerning the auroral variations in Fig. 4 and Fig. 8, LOS speeds
show diffusion during auroral events. Negative LOS speeds initially concentrated northward, spread westward and eastward,
expanding horizontal coverage. Positive LOS speeds initially in the southward direction shrink. When converted to horizontal
wind speeds, these changes lead to increased horizontal differences, or the false convergence caused by scattering, in other



words. The northward LOS speed changes slightly, the southward speed changes the most and nearly reverses, while the
eastward and westward speeds are intermediate.

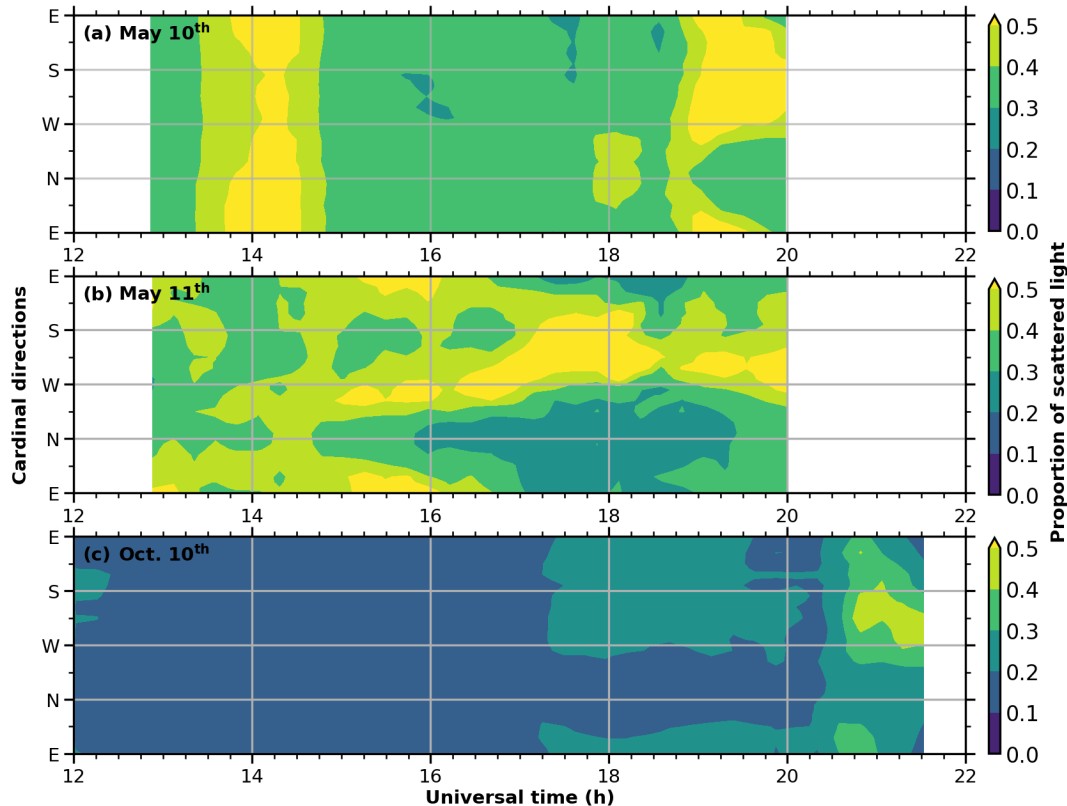


**Figure 8: Proportion of scattered light at 45° zenith angle**
The proportion is the ratio of scattered light intensity to the total light intensity (both scattered and direct) in the simulation. The figure
shows this proportion at 45° zenith angle for various directions over time, with panels for May 10[th], May 11[th], and October 10[th],
respectively.
Figure 8 shows the ratio of scattered light intensity to total light intensity at a fixed 45° zenith angle calculated by the scattering
model. Consistent with the schematic diagram (Fig. 1), the scattered intensity is the sum of all injected directions, and the total
light intensity includes the direct component on this basis. Without auroras, the scattering proportion is typically below 0.4
and varies with atmospheric scattering capability. However, during auroral events, the scattering proportion in some directions
can increase to 0.5 or higher. The northward scattering proportion increases the least and remains much lower than in other
directions. In contrast, the scattering proportion is significantly enhanced in directions ranging from 135° to 180° away from
northward. The aurora appears in the north, resulting in much higher northern brightness. After atmospheric scattering, light
from the north diffuses into surrounding directions, increasing the scattering proportion. Because the north itself has strong



direct light, its scattering proportion remains small. In the model, we assume that without considering temperature-related
spectral broadening, stronger light rays dominate interference fringe identification (Wei et al., 2020), thereby determining the
Doppler shift or LOS speeds. The lower scattering proportion in the north allows northward observations to retain more LOS
speeds from themselves, while other directions experience greater LOS speed contamination from the north. Therefore, the
northward simulation is closest to the default inputs, the southward simulation deviates the most, and the eastward and
westward simulations lie in between. Additionally, the scattering impact should also consider the initial LOS speeds in the
brighter region. Despite the high scattering proportion on May 11th shown in Fig. 7, the simulated LOS speed changes in Fig.
8 are minimal. This is because of the smaller LOS speed in the auroral region on that day, resulting in less contamination
spread to other directions.
The core working principle of the scattering model relies on the relationship between airglow brightness and background LOS
wind speeds. As Harding (2017a) noted, scattering requires a bright sky region with large LOS wind speeds. Firstly, spatially
uneven airglow brightness is a prerequisite for significant scattering. The brightest airglow area contributes most to scattered
light intensity, and its Doppler shifts determine the LOS biases in other directions. This principle allows a rough assessment
of scattering impact without model computation when airglow is uneven. Locate the brightest region and its Doppler shift type,
as the same Doppler shift type will likely appear in other directions. Blue-shift dominance indicates increased downward wind
and horizontal deviations opposite to the line of sight, resembling convergence, while red-shift dominance resembles
divergence. Minimal scattering-induced biases occur if the scattering proportion is very low due to uniform airglow brightness,
or if the LOS velocity in the bright region is near zero (i.e., the wind speed is perpendicular to the line of sight). Previous
observations can be directly verified by this principle and are basically in line with it (Hernandez and Roble, 1976; Price et al.,
1995; Ishii et al., 1999; Ishii et al., 2001; Makela et al., 2014). Unfortunately, scattering impact can complicate dynamic
analysis. In polar regions, auroras are characterized by green-line emissions, and thermospheric winds are significantly
influenced by ion drag, where scattering effects may not be prominent. In contrast, mid-latitudes have mainly red-line auroras
with large-scale uniform circulation, making the scattering impact more pronounced and distinguishable.
**4.2 Errors of the scattering model**
The model also exhibits certain errors and limitations. Scattering-induced biases in observations have nearly similar
magnitudes on May 10th and Oct. 10th. However, with the same 100 m·s⁻¹ westward and 400 m·s⁻¹ southward wind input, the
scattering-induced biases in the May case are significantly larger in magnitude and closer to reality compared to October. In
Fig. 7, the simulated LOS speeds show a larger diffusion range for the May 10th case compared to October. In Fig. 8, the
scattering proportion for May 10th is consistently higher than that for October. We attribute this difference to the distinct optical
depths in the two months, which are 0.43 and 0.2, respectively. Optical depth reflects atmospheric extinction capability and is
mainly related to aerosol content (see Appendix B). It primarily affects the extinction process and influences the magnitude of
scattering-induced biases by altering the proportion of scattered light. When the optical depth is artificially increased to a





higher value, such as around 0.6, the model more closely matches the Oct. observations. We find that the model underestimates
scattering effects when the optical depth is low.
In the October 10[th] event, the simulated scattering-induced biases are inconsistent between the two auroral brightness peaks.
In Fig. 7, the LOS speed variation is larger during the second peak, and in Fig. 8, the scattering proportion is greater. This is
because the scattering proportion is susceptible to errors in scattered and direct light intensities. DCAI doesn't correct for
Rayleigh units, leading to significant errors in regions with large zenith angles. The model can't fully eliminate the stray light
caused by the glass dome when separating the initial direct and scattered light from DCAI images (Harding et al., 2017a),
resulting in errors. In our experiments, if the stray light effect isn't subtracted as described in Appendix B, the model becomes
more inert, resulting in a smaller simulated scattering proportion.
In our experiments, the scattering model consistently underestimates scattering impact compared to observations, despite
several applied enhancements: (1) A single-scattering albedo of 1 was used, ignoring absorption. (2) Stray light effects were
removed. (3) Attenuated airglow observations at the edge of DCAI images were extrapolated, enhancing edge scattering. (4)
Excessive edge extinction was reduced by correcting the extinction path geometry, increasing the scattered light intensity
integral. (5) Zero vertical wind was assumed when converting simulated LOS speeds to geographical wind speeds. Since the
integral only includes 10 optical depth layers, with each light ray scattering once per layer and extinguishing once between
layers, it may be too crude compared to the real path, underestimating the scattered light. Simply increasing the number of
optical depth layers is not effective. We think this may be related to the non-linear variation of atmospheric density with
altitude, where optical depth may not vary linearly with height, and the scattering phase function may also change with altitude.
To address this issue, future work should complete the DCAI correction. Additionally, introducing a model of optical depth
varying with altitude can increase the number of single-scattering nodes and ensure the geometric accuracy of the extinction
path, thereby improving the accuracy of scattered light intensity calculations.
Furthermore, these bright region observations do not necessarily correspond to the wind speeds of the red-line airglow layer.
They could also come from higher auroral layers and not even reflect the neutral thermospheric winds near 250 km (Makela
et al., 2014). In Fig. 3 and Fig. 5, the north-looking wind observations show unusually high wind speeds, which are significantly
different from the simulations. In particular, on October 10[th], the north-looking wind speed varied dramatically with the
intensity of the northern aurora. During the two auroral peaks, the north-looking wind direction also reversed. The north-
looking wind speed changed more than the south-looking wind speed, inconsistent with the simulation characteristics. When
auroras occur, the north-looking wind speed in the bright region may be significantly higher than the average value over zenith
and distinct from regions outside the aurora. This could enhance scattering effects, which the model fails to simulate.



### 4.3 Scattering effects on thermospheric wind observation

Given the directional inhomogeneity of scattering impact, the reliability of the commonly utilized average wind and individual direction wind observations needs to be considered. Wind speed detected from a less bright region is more doubtful, because these regions are more polluted by scattered light. The average speed of opposite cardinal directions will maintain the correct time-varying trend of the background thermospheric wind but may deviate in absolute value. Since the LOS speed deviations caused by scattering in each direction have the same type, averaging can cancel out the scattering deviations between opposite directions, making the average wind closer to the true background thermospheric wind. However, in directions with strong light intensity gradients, such as north-south, the deviations in the two directions are different, and averaging will cause the wind speed to deviate from the true value. Additionally, the wave-like structures appearing in each individual direction may originate from changes in scattering effects caused by variations in auroral intensity, and whether they come from TADs needs further verification. The wave-like structures superimposed on the average value may also have their amplitudes enhanced or cancelled due to the directionality of scattering deviations, which similarly requires additional validation.

### 5 Conclusions

This study has further proved that lower atmospheric scattering can bias thermospheric wind observation on ground-based optical interferometers. The light scattered from the non-line-of-sight directions of the scope will lead to additional LOS speeds and appear as atypical horizontal differences and vertical wind at geographic coordinates. With a simplified scattering radiative transfer model, we simulate the distribution of airglow intensity after the multiple scattering of the lower atmosphere and estimate the wind observation bias under scattering impact via a weighted average method. Atypical winds under conditions of spatially uneven airglow have been generally captured.

We have refined the scattering model in previous research to enhance its computational efficiency. Specifically, we simplified the LOS wind speed simulation and introduced an upper limit for the lower atmosphere to improve the accuracy of the extinction length calculation. The scattering impact can be directly estimated through the relationship between the bright airglow region and the LOS wind speed. The brightest airglow region contributes most to the scattering impact, of which the Doppler shift type determines the LOS biases in other directions. Due to the directional inhomogeneity of scattering effects, we propose that in directions with strong light intensity gradients, such as north-south, the average wind speed of opposite directions and its fluctuations may be biased. Although the observed winds are affected by scattering when airglow is uneven, they still retain dynamic information, such as the average wind being close to the storm-time circulation. Unfortunately, we lack alternative observational methods to verify the accuracy of the interferometer results. It deserves further study to the extent of scattering impact with more cases and additional instrumental observations.



Limited by the accuracy of the model inputs, the scattering model can only simulate the wind features associated with scattering
impact under clear sky conditions. It remains incapable of precisely picking out the speed alterations induced by scattering
impact. Given that scattering significantly impacts observations during geomagnetic storms accompanied by auroras, it is
necessary to quantify these biases to provide accurate data for dynamic studies. Therefore, future efforts can focus on refining
the model and its inputs or statistically analyzing scattering biases under different airglow brightness distributions from various
stations to gain a clearer understanding of the magnitude of scattering biases.
**Appendix A**
In the following appendices, we provide a concise description of the scattering model's operational principles, inputs, and
modifications employed in our works. For more detailed solutions, please refer to the article by Harding et al. (Harding et al.,
2017a).
Based on the radiation transfer theory, Hansen and Travis (1974) and Sobolev (1975) gave the multiple scattering solution.
Harding (2017a) extended the initial source function $J_0(\tau, u, \phi)$ to airglow surface source, and corrected the missing
normalization factor in the scattering phase function (Eq. (1) to Eq. (3)):
$$u\frac{\mathrm{d}I(\tau,u,\phi)}{\mathrm{d}\tau} = -I(\tau,u,\phi) + J(\tau,u,\phi) \tag{1}$$

$$J(\tau,u,\phi) = \frac{\omega}{4\pi}\int_0^{2\pi}\int_{-1}^{1} P(u,u',\phi,\phi')I(\tau,u',\phi')\mathrm{d}u'\mathrm{d}\phi' + J_0(\tau,u,\phi) \tag{2}$$

$$J_0(\tau,u,\phi) = \frac{\omega}{4\pi}\int_0^{2\pi}\int_0^{1} P(u,u',\phi,\phi')\sec(\gamma')f(u',\phi')\exp[-\tau L(u')]\mathrm{d}u'\mathrm{d}\phi' \tag{3}$$

$$L(u) = [(R_e + H_L)\cos(\gamma_\tau) - R_e u]H_L^{-1} \tag{4}$$

$$u = \cos(\zeta) \tag{5}$$

The equations are formulated within an improved local spherical coordinate system, including azimuth $\phi$, zenith angle $\zeta$
which is represented in cosine form $u$, and vertical height which is expressed as optical depth $\tau$.
In the case of scattering, as illustrated in Fig. 1, the light intensity along a line of sight, represented by $I(\tau, u, \phi)$, consists of
two parts, the direct light (a in Fig. 1) from the same direction, and the aggregate of scattered light (b-e in Fig. 1) from other
directions, which is represented by the source function $J(\tau, u, \phi)$. Based on the radiative transfer equation (Eq. (1)), at each
optical depth layer, the scattered light intensity received from all directions will be integrated. Simultaneously, the original
intensity in the line of sight will be added to the total scattered light. Besides, the extinction in the path will be calculated
according to the optical depth.



There are two potential scattering paths in the lower atmosphere: single scattering (b, c in Fig. 1) and multiple scattering (d, e
in Fig. 1). The model computes them sequentially via an iterative process. In the initial state, there is no light intensity in the
lower atmosphere. Therefore, the single scattering will originate solely from the airglow layer, and the source function
$J(\tau, u, \phi)$ will be equivalent to the initial source function $J_0(\tau, u, \phi)$ at this state. By solving Eq. (1), the model can obtain
the single scattered intensity in each direction at every optical depth layer, which is the updated source function $J(\tau, u, \phi)$.
Then, the multiple scattering can be calculated based on it. Typically, the total scattered intensity will remain relatively constant
when accounting for the fourth scattering. By using this iteration, the scattered light and residual direct light in DCAI images
can be effectively separated. The residual direct light will subsequently serve as the background intensity distribution for
simulating speeds.
In the source function $J(\tau, u, \phi)$, the scattering phase function $P(u, u', \phi, \phi')$ quantifies the relative gain of an incident angle
to an exit angle during the scattering process. The reference value is based on a unit-radius sphere, which necessitates the
introduction of a factor $\frac{1}{4\pi}$. Furthermore, ω represents the single-scattering albedo, set as 1. The initial source function
$J_0(\tau, u, \phi)$ is responsible for introducing the airglow distribution $f(u', \phi')$. Here, $\sec(\gamma')$ represents the secant of the zenith
angle at the puncture point of the airglow layer, which helps eliminate the Van Rhijn effect. Additionally, the exponential term
with base e is utilized to calculate the equivalent extinction length along an inclined path.
It is primarily the lower atmosphere that significantly scatters and absorbs light (He et al., 2021; Li et al., 2022). Therefore,
when computing the extinction length, just employing the cosine of zenith angle $u'$ will lead to an overestimation of the
effective length, as illustrated on the right side of Fig. 1. To address it, we set an upper boundary $H_L$ of the lower atmosphere
at 40 km, assuming an optical depth of zero above this altitude. Using geometric relationships, an equivalent length factor
$L(u)$ can be derived, where $R_e$ means Earth radius, and $\gamma_\tau$ represents the zenith angle at the penetration point of 40 km
height. This value can be readily calculated by adjusting the target height of the formula for $\gamma$. Inside the lower atmosphere,
we apply a thin-layer approximation, which also utilizes the geometric relationships at the upper boundary.
$$V = \frac{V_{dr}I_{dr} + \sum_k V_{sc}(k)I_{sc}(k)}{I_{dr} + \sum_k I_{sc}(k)}$$
(6)

After working out the background intensity distribution, we partition the LOS speeds at 250 km into several bins. Ignoring the
background temperature gradient, LOS speeds can directly correspond to Doppler shifts to simplify the simulation of the
Doppler distribution. As roughly illustrated in Fig. 1, all LOS speeds are categorized into k=10 bins valued from highest to
lowest, assuming that the LOS speeds within each bin approximate their mean value $V_{sc}$. The scattered intensity distribution
$I_{sc}$ is computed by extracting the airglow brightness from the corresponding region of each bin. And, there will be no intensity
from other areas during a single bin's computing. According to Eq. (6), the simulative LOS speed at a specific angle will be
an averaged result, where the original speed $V_{dr}$ is weighted by the direct light intensity $I_{dr}$, and the additional speed resulting



from scattering $V_{sc}$ is weighted by the scattered light intensity $I_{sc}$. The model directly uses the average within DCOI's 9
degree field of view as simulated post-scattering LOS speed of interferometers, since DCOI and FPI haven't measured the
reception gain of light outside their fields of view. We find that due to the coarse model grid, the 9 degree average is nearly
the same as using the nearest single-sight observation. Finally, the LOS speed will be converted to horizontal wind, maintaining
the assumption of zero vertical wind to prevent the propagation of scattering biases in the vertical direction.

**Appendix B**

This appendix details the scattering model's inputs from measurements, supplementing the second section. The background
airglow brightness for the model comes from DCAI. Image processing includes: (1) dark field exclusion, (2) median filtering
to remove starlight, (3) conversion to the local spherical coordinate, (4) stray light correction, (5) radial zero-order
extrapolation for regions beyond 70° zenith angle. Stray light results from the scattering of strong incident light by the glass
dome. During quiet nights without auroras, it is weak and uniformly distributed across all LOS directions. We use the azimuthal
average of the nearest quiet night at 45° zenith angle as a reference for weak stray light condition. After aurora onset, stray
light brightens all directions. The difference between the darkest direction at 45° zenith angle and the reference value is
considered the additional stray light caused by the aurora and is subtracted from the entire image. The shown airglow images
additionally mitigate the Van Rhijn effect through $\sec(\gamma')$ and the atmospheric extinction through $\exp[-\tau L(u')]$ (see
Appendix A). Since the scattering model already includes these processes, no separate treatment is needed. The optical depth
and scattering phase function inputs are shown in Fig. 9. Optical depth is calculated using AERONET's monthly averages,
with interpolation at 630 nm. Since only daytime observations are available, local daytime values are used to represent
nighttime values. Fig. 9a and 9b show monthly average optical depths at local daytime, with total averages of 0.43 and 0.2.
The scattering phase function is a weighted average of molecular and aerosol scattering phase functions from AERONET at
675 nm, which is weighted on the total optical depth of aerosols and molecules.



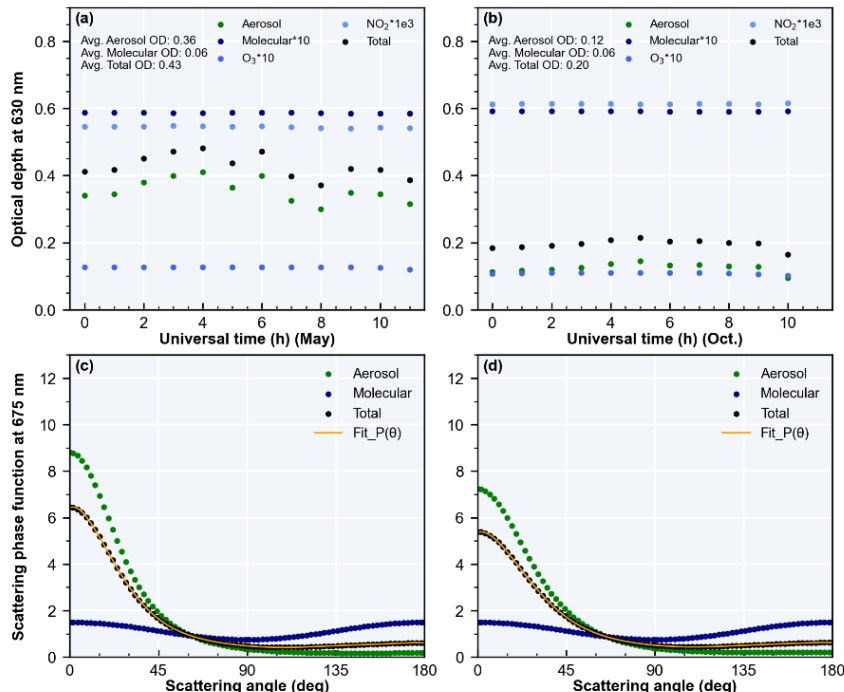

**Figure 9: Optical depth and scattering phase function in May and October**

The first two panels show daytime monthly average optical depths, with molecular optical depth amplified for clarity. The rest panels show the scattering phase functions.

**Code availability**

The code of DCAI images correction, the scattering model, and the visualization are not publicly available yet. If needed, they can be obtained by contacting the corresponding authors via email.

**Data availability**

The data of DCOI and DCAI from the Chinese Meridian Project can be obtained from https://www.meridianproject.ac.cn/en/. The FPI data can be obtained by contacting the corresponding authors via email. The data of AERONET can be obtained from https://aeronet.gsfc.nasa.gov/. The Kp index provided by GFZ, German Research Centre for Geosciences, can be obtained from https://kp.gfz-potsdam.de/en/.



**Author contributions**

Conceptualization: XW; investigation: XW, GJ; instruments construction and maintenance: YZ, JX, WL, TW, GZ, WY; wind data retrieval: YZ, WL, GZ, TW; airglow images correction: XW; model improvement and programming: XW; visualization: XW; validation: GJ, YZ, JX, WL, TW; writing - original draft preparation: XW; writing-review and editing: GJ, YZ, JX, WL, TW; supervision: GJ, YZ; project administration: GJ, YZ; funding acquisition: YZ. All authors contributed to the revision and improvement of the paper.

**Competing interests**

The authors declare that they have no conflict of interest.

**Acknowledgements**

We appreciate all the funding from the National Key R&D program of China (2023YFB3905100), the Project of Stable Support for Youth Team in Basic Research Field, CAS (YSBR-018), the National Natural Science Foundation of China (42174212), the Chinese Meridian Project, and the Specialized Research Fund for State Key Laboratories. We acknowledge the use of data from the Chinese Meridian Project. We thank all the builders and maintainers of DCOI, FPI, and DCAI of Siziwang station. We thank Lingli Tang for their effort in establishing and maintaining AOE_Baotou site of AERONET.

**Financial support**

This work was supported by the National Key R&D program of China (2023YFB3905100), the Project of Stable Support for Youth Team in Basic Research Field, CAS (YSBR-018), the National Natural Science Foundation of China (42174212), the Chinese Meridian Project, and the Specialized Research Fund for State Key Laboratories.

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
