# Peer review of "Impact of lower atmospheric scattering on ground-based optical"

_EGUsphere, 2025_

## Author Comment (AC1)

Dear Referee #2,

Thank you very much for your thoughtful review and constructive comment. Following your earlier comment on the red- aurora issue, we still lack the observations required for a definitive analysis, and the TIE-GCM-based estimate indeed has uncertainty. At this stage, we are unable to resolve this issue. Through our discussions with you, we realized that relying solely on qualitative pattern recognition to emphasize scattering while overlooking other mechanisms was premature. Accordingly, we have extensively revised the Introduction and Discussion to (1) remove the previous qualitative inference in the Introduction, emphasize that scattering is only one of several possible causes of the wind differences, and clearly state our current limitations, (2) add relevant background and discussion on red aurora and (3) remove the incomplete assessment of scattering's impact on wind observations from the Discussion. Thank you for your insightful comments. It has led us to conduct more comprehensive and rigorous consideration and discussion. The following is our thinking and modification.

Our research is based on the spatially uneven airglow. At mid-latitudes, the main factor significantly causing uneven red-line airglow is the red aurora. Airglow and aurora form differently, but their emission bands and height profiles are so alike that ground optical instruments record them as one. Red aurora, mainly from <80 eV electrons, produces 630.0 nm stable auroral red arc (SAR) in mid-latitudes during storm recovery (Upadhyay et al., 2025) and has also been observed in the main phase (Shiokawa et al., 2013). Red aurora is thought to inject less energy and to spread over a broader region than auroras at other wavelengths (Rees and Luckey, 1974; Gabrielse et al., 2021). Kataoka et al. simulated red-line profiles for comparison between

May 9th and May 11th (Kataoka et al., 2024). They found that the red aurora lifted the emission peak above 300 km and markedly increased the intensity above the peak. Consequently, when the interferometer faces the aurora, it may sample winds from higher and farther away than the usual 250 km height. We agree that such errors could exist, and the significant deviations in the northward observations and scattering models seem to point to this as well.

In our last response, TIE-GCM was taken as a large-scale storm-time wind field for reference. We used it to estimate that the wind difference caused by the interferometer's shifted sight would probably be smaller than what we observed. About the TIE-

GCM simulation, we agree that its spatial resolution and empirical convection pattern could not accurately catch small-scale aurora (Matsuo and Richmond, 2008). It primarily simulates large-scale Joule heating processes in the polar region and the resulting thermospheric surge. Using the TIE-GCM to address this issue is indeed likely to introduce considerable uncertainty.

At present, we lack additional thermospheric-wind or auroral instruments poleward of our station, so the true value of both the red aurora and the neutral wind remains unknown, and the possible influence of red aurora cannot yet be ruled out.

In the original Introduction, we relied on a qualitative assessment of the observed pattern and ascribed the wind bias primarily to scattering. We now realize that this conclusion was premature. It could mislead readers into ignoring other possible causes.

Therefore, we have carefully revised the Introduction to clarify that, among dynamical processes, red-aurora effects, scattering, and other factors, only the scattering can be evaluated with our model, whereas the others cannot yet be ruled out. We have removed the assessment of scattering's impact on wind observations from the Discussion (Section 4.3), since we overestimated the scattering impact on red-line wind measurements in the previous manuscript. Our observations likely contain complex contributions from aurora and dynamics that, as noted, cannot yet be quantitatively separated. Section 4.3 has therefore been removed to prevent an incomplete assessment. We have also revised several related yet overly assertive statements elsewhere in the manuscript. In addition, we have added brief context on the aurora effects in the Introduction and expanded the

Discussion to address non-scattering influences in greater detail. Below, we attach the key revisions. All corresponding revisions in the manuscript are highlighted in purple. Thank you again for this valuable suggestion.

**The key revisions are:**

**(1) Add a paragraph on aurora and other potential influences at L59 of the Introduction.**

"Scattering-induced biases are more pronounced during spatially uneven airglow brightness, such as during auroras (Harding et al., 2017a). Uneven airglow brightness refers specifically to inhomogeneous red-line emissions. At mid-latitudes, marked uneven red-line airglow usually comes from red aurora. Despite their distinct origins, the spectral and altitudinal overlap of airglow and aurora will let ground-based optical instruments conflate the two. **For red-line observations, the aurora itself**

**may also bias the derived winds.** Aurora could elevate the red-line emission profile (Kataoka et al., 2024b), so the interferometer samples winds that are both higher and farther away. This makes the northward view sense winds deviate from the expected thermospheric wind at 250 km altitude when looking toward the aurora. Additionally, spectral contamination from precipitating energetic ions could also bias interferometers (Makela et al., 2014). They suggested that the enhanced downwelling at mid-latitudes during storms might result from the contamination of the spectral profile by fast O atoms associated with the influx of low-energy $O^+$ ions."

**(2) We have revised the end of the Introduction to remove any assertive qualitative analysis and to state the objectives**

**of our study more clearly.**

"During two geomagnetic storms on May 10th and October 10th, 2024, …… These atypical winds at SIZW only occurred with auroras statistically and significantly deviated from the regional climatological norms over the China region (Jiang et al.,

2018; Yang et al., 2020). This raises the question of whether the atypical winds arise from dynamical processes, are influenced by red aurora, or stem from scattering-induced biases and other measurement-related factors. Unfortunately, most of these mechanisms could amplify the wind-speed contrast between opposite cardinal directions, rendering them difficult to disentangle (Harding et al., 2017a).

**Given the scarcity of additional thermospheric-wind or**

**auroral instruments, we remain unable to quantify every potential mechanism. Motivated by the observed phenomena,**

**this study attempts to estimate how scattering modulates the atypical winds in these storms.** While prior studies focus on vertical wind biases of Fabry-Perot interferometers under auroral conditions (Harding et al., 2017a; Harding et al., 2017b), we will analyze the formation and patterns of horizontal differences caused by scattering. We will also incorporate Doppler

Asymmetric Spatial Heterodyne (DASH) interferometer data to compare scattering impact across different interferometer types. As red auroras now regularly appear at the low magnetic latitudes of Japan and China during elevated solar activity (Kataoka et al., 2024a; Kataoka et al., 2024b; Ma et al., 2024), a deeper understanding of scattering-induced biases is essential for the proper use of interferometer data collected in these regions. In the following text, a scattering radiative transfer model is used to simulate interferometer observations in two cases with visible aurora. The presence and patterns of scattering-induced biases are analyzed by comparing simulations with observations."

**(3) We have revised L405–412 of the Discussion to expand the auroral influences.**

"Furthermore, these bright region observations do not necessarily reflect the usual 250 km thermospheric wind. In Fig. 3 and

Fig. 5, the north-looking wind observations show unusually high wind speeds, which are significantly different from the simulations. In particular, on October 10th, the north-looking wind speed varied dramatically with the intensity of the northern aurora. During the two auroral peaks, the north-looking wind direction also reversed. This indicates that the interferometer receives an additional effect when it looks toward the aurora. **This indicates that the interferometer receives an additional**

**effect when it looks toward the aurora.** Kataoka et al. showed that red aurora lifted the red-line emission profile, raising its peak above 300 km and brightening the upper part on May 11[th] (Kataoka et al., 2024b). Consequently, the interferometer can sample winds that are higher and more poleward. Because storm-time surges propagate from the polar region to the equator, these higher, poleward regions are likely to carry stronger equatorward winds. The interferometer may record a larger wind speed toward the aurora. Additionally, spectral contamination from precipitating energetic ions can also bias interferometers (Makela et al., 2014). In other words, the interferometer is partly sensing the speed of non-neutral species, boosting the observed wind. These issues lie beyond what scattering models can reproduce. From the observed pattern, we infer the presence of non-scattering effects, especially in the poleward view. Due to the absence of nearby higher-latitude neutral-wind observations relative to SIZW, quantifying their respective contributions remains challenging."

**(4) Section 4.3 has been removed from the discussion since we overestimated the scattering impact on red-line wind**

**measurements in the previous manuscript.**

**References:**

Gabrielse, C., Nishimura, T., Chen, M., Hecht, J. H., Kaeppler, S. R., Gillies, D. M., Reimer, A. S., Lyons, L. R., Deng, Y.,
Donovan, E., and Evans, J. S.: Estimating Precipitating Energy Flux, Average Energy, and Hall Auroral Conductance
From THEMIS All-Sky-Imagers With Focus on Mesoscales, Front Phys-Lausanne, 9, ARTN 744298,
doi:10.3389/fphy.2021.744298, 2021.

Kataoka, R., Reddy, S. A., Nakano, S., Pettit, J., and Nakamura, Y.: Extended magenta aurora as revealed by citizen science,
Scientific Reports, 14, ARTN 25849, doi:10.1038/s41598-024-75184-9, 2024.

Matsuo, T. and Richmond, A. D.: Effects of high-latitude ionospheric electric field variability on global thermospheric Joule
heating and mechanical energy transfer rate, J Geophys Res-Space, 113, Artn A07309, doi:10.1029/2007ja012993, 2008.

Rees, M. H. and Luckey, D.: Auroral Electron-Energy Derived from Ratio of Spectroscopic Emissions .1. Model Computations,
J Geophys Res, 79, 5181-5186, doi:10.1029/JA079i034p05181, 1974.

Shiokawa, K., Miyoshi, Y., Brandt, P. C., Evans, D. S., Frey, H. U., Goldstein, J., and Yumoto, K.: Ground and satellite
observations of low-latitude red auroras at the initial phase of magnetic storms, Journal of Geophysical Research: Space
Physics, 118, 256-270, doi:10.1029/2012JA018001, 2013.

Upadhyay, K., Shiokawa, K., Pallamraju, D., and Gololobov, A.: Determination of electron heat flux in the topside ionosphere
and its impact on the vertical profile of OI 630.0 nm emission rate during nighttime SAR arcs for different solar activity
conditions, Advances in Space Research, 75, 4731-4739, doi:10.1016/j.asr.2024.12.046, 2025.

---

## Author Comment (AC2)

Dear Referee #1,

We are deeply grateful for your thoughtful review. Your suggestions and insights have greatly improved our work. We have addressed every comment and detailed our responses below. Following your suggestions, we have revised the Introduction, the scattering-model description, and the Discussion to improve clarity and accuracy. All corresponding revisions in the manuscript are highlighted in purple.

We note your expectation that wind-speed fluctuations should decrease from U_LOS_S to U_LOS_Z to U_LOS_N. Our simulation reproduces this order while the observations do not. Below, we want to clarify the scattering mechanism and discuss the remaining simulation–observation discrepancy briefly.

The post-scattering speed change is indeed direction-dependent, **as detailed in Section 4.1, "Core working principle of the**

**scattering model"**. There, we show the simulated horizontal LOS speed variations (Fig. 7 of the manuscript) together with the proportion of scattered light (Fig. 8). In short, the wind-speed fluctuation for any direction depends on (1) its geometry relative to the brighter airglow patch and (2) whether its LOS speed sign matches that of the brighter patch.

Unfortunately, simulation–observation discrepancies remain: (1) the model underestimates scattering-induced deviations in all directions (case of Oct.), and (2) the north-looking winds are unexpectedly larger than simulated and depart from the expected

U_S > U_Z > U_N ordering (case of May). These discrepancies point to either model underestimation of scattering or extra non-scattering effects. We have refined the model on multiple aspects to reduce its underestimation and explored possible non- scattering effects. However, with no ISR or other neutral-wind measurements available in the study region, quantifying these non-scattering contributions remains hard. **We have summarized the possible influencing factors in Section 4.2, "Errors**

**of the scattering model"**. In short, for discrepancy (1), only optical depth tuning has so far appreciably reduced the model's underestimation. When the optical depth is artificially increased to a higher value, the model outputs more significant wind- speed fluctuations and more closely matches the Oct. observations (0.2 in practice). The optical depth used in the manuscript is a daytime observation obtained ~180 km from Siziwang, and thus carries substantial uncertainty. The figure below shows the simulated results for Oct. 10[th] under different optical depths.

[Figure]

For discrepancy (2), we suspect the north-looking line of sight picks up non-neutral Doppler shifts from embedded auroral emissions. **That is discussed at the end of Section 4.2.** Because of multiple uncertainties in the input fields, the simulation can hardly match the observations numerically at present. Nevertheless, it reproduces the observed wind-speed trends and is therefore useful for examining the scattering mechanism. For now, we can only list and discuss the potential issues that arise between the model and the observation. A quantitative verification will require additional aerosol and radio measurements that await future work.

**Below are our responses:**

**Comment #1:** *L70: "During two geomagnetic storms with visible auroras" > "During two geomagnetic storms on May 10th*

*and Oct. 13, 2024 with visible auroras"*

**Response #1:** Thank you for this helpful suggestion. We feel sorry for not specifying the storm dates in the introduction. We have revised **L70 to: "During two geomagnetic storms on May 10th and October 10th, 2024, with visible auroras".**

**Comment #2:** *L72-73: "The observations were unaffected by moonlight or clouds, and the interferometer retrieval errors*

*were acceptable." How did you confirm these?*

**Response #2:**

To keep the Introduction concise, we deferred the complete data-screening criteria to Section 3.1, L177–180. Thank you for highlighting the potential confusion. We have now added a forward reference **at L73 "…(see Section 3.1)"** to guide readers directly to these details.

The screening criteria are (1) excluding cases where the angle between the moon and the line of sight is less than 30 degrees, (2) excluding cases where large-area thick cloud coverage is visible in DCAI, and (3) excluding data with standard errors greater than 50 m·s⁻¹. We computed the moon's position with PyEphem (https://rhodesmill.org/pyephem/index.html). As the cloud sensor at Siziwang was not yet operational in 2024, we reviewed the all-sky imager DCAI and excluded any intervals where excessively thick clouds obscured the field of view.

**Comment #3:** *L139: "The LOS speed is used directly instead of the Doppler shift, assuming a constant background*

*temperature." This sentence requires revision. I have two comments: (1) To my knowledge, the LOS speed is derived from the*

*Doppler shift measured by the optical interferometer. In this process, the Doppler shift seems more like a "direct" measurement*

*rather than the LOS speed itself. (2) I find it unclear why the assumption of a constant background temperature is necessary.*

*This should be more explicitly addressed in the text.*

**Response #3:**

Thank you for the insightful suggestion. With the instrument thermal drift kept under control and elastic scattering assumed, we skip the chain fringe-shift > frequency-shift > LOS speed and simulate each ray directly with its LOS speed. The Doppler shift remains implicit and need not be computed explicitly. Every incident ray from the airglow layer is mapped straight to its

LOS speed. This approach is outlined in Response #4.

However, during storms, the thermosphere heats rapidly, broadening the emission line and increasing the retrieval uncertainty.

We are concerned that spatial temperature gradients near the auroral zone may further enlarge the deviation between simulation and observation. Thus, we highlighted the limitation in the previous manuscript. Indeed, wind-speed retrieval itself is insensitive to thermosphere temperature. The latter merely affects the retrieval uncertainty. We have now revised the relevant statements to minimize any possible misinterpretation:

**L139:** "(2) The LOS speed is used directly instead of the Doppler shift, assuming a constant background temperature." **to "(2)**

**The Doppler shift is replaced by LOS speed, with every incident ray from the airglow layer mapped directly to its**

**corresponding LOS speed."**

**Remove L141-145:** "We directly use LOS speed instead of Doppler shift, primarily neglecting the interference fringe recognition errors caused by spectral broadening due to temperature variations. During auroral events, FPI observations show similar neutral temperatures in all directions, with the northward direction occasionally being about 300 K higher (not shown here). Overall, the temperature at mid-latitudes is uniform at the 500 km spatial scale, and the variations caused by spectral line broadening can be neglected."

**Background temperature is added to Discussion (4.2) as a possible source of error:**

**"We also considered the potential influence of thermospheric temperature. FPI data show uniformly elevated**

**thermospheric temperatures in these two storms, with the northward view occasionally about 300 K warmer than the**

**others (not shown here). Because our scattering model does not yet include temperature effects, we cannot quantify**

how much scattering biases the FPI temperature measurements. In the study of Harding et al. (2017b), wind simulations are temperature-independent, while temperature retrieval relies on the wind. Likewise, we substitute the LOS speed for the Doppler shift and ignore temperature-induced spectral broadening. In principle, thermospheric temperature influences retrieval uncertainty, not the wind speed itself. We remain cautious that ignoring this uncertainty could introduce extra bias if a horizontal temperature gradient is present, but incorporating it would markedly raise the computational cost and remains a task for the future."

**L352-353:** "In the model, we assume that without considering temperature-related spectral broadening, stronger light rays dominate interference fringe identification" **to "In the model, we assume that stronger light rays dominate interference fringe identification"**

**L491-493:** "Ignoring the background temperature gradient, LOS speeds can directly correspond to Doppler shifts to simplify the simulation of the Doppler distribution." **to "To simplify simulation, the model directly uses LOS speeds corresponding to Doppler shifts."**

**Comment #4:** *L140: "After binning different LOS speeds and computing the corresponding scattered light intensity, contaminated LOS speeds are calculated via weighted average, simplifying the wind simulation." The procedure for "binning different LOS speeds" is unclear to me. Is it similar to pixel binning, a technique used in digital camera sensors?*

**Response #4:**

This is different from pixel binning. The specific method is described at the end of Appendix A (L490–498). In short, (1) We converted the assumed horizontal wind to the LOS speed at each viewing angle on the airglow layer, ignoring the effect of earth curvature. (2) We slice the airglow layer by LOS speed. Each bin spans less than ±40 m s$^{-1}$. We simply take its mean speed $V_{sc}$. (3) At each run, we illuminate a single bin to simulate the scattered intensity $I_{sc}$ seen from all angles on the ground (all rays share that bin's LOS speed). (4) At each viewing angle, the post-scattered LOS speed is computed as an intensity-weighted average (including both direct ($V_{dr}, I_{dr}$) and scattered components) according to the following equation:

$$V = \frac{V_{dr}I_{dr} + \sum_k V_{sc}(k)I_{sc}(k)}{I_{dr} + \sum_k I_{sc}(k)}$$

Because the detailed method is given in the appendix, the main-text description may appear too vague. We have now made it more specific:

**L140-141:** "(3) After binning different LOS speeds and computing the corresponding scattered light intensity, contaminated LOS speeds are calculated via weighted average, simplifying the wind simulation." **to "(3) After slicing the airglow layer into several bins by LOS-speed, the model illuminates one bin per run, records its scattered intensity, then merges all bins with an intensity-weighted average to yield the post-scattered LOS speed."**

---

## Author Response (AR2)

- 1 Dear Referee #1,
- 2 Thank you for your helpful comment. Previously, constrained by simulation uncertainties, we avoided numerical specifics and
- 3 emphasized qualitative descriptions. Adding quantitative detail indeed helps readers grasp the model's performance and
- 4 limitations more intuitively. While preserving scientific rigor, we have accordingly inserted additional quantitative information
- 5 on the scattering simulation wherever possible.
- 6 These additions include:
- 7 (1) The magnitude of simulated scattering biases (horizontal differences);
- 8 (2) The relative changes in LOS speed among different directions;
- 9 (3) The relative changes in simulated scattering biases under different experimental conditions.
- 10 These revisions are concentrated in the Abstract, Discussions, and Conclusions, with additional supporting statements added
- 11 to the Results. Below we list the key revisions, highlighted in purple.

**12 Comment:**

- 13 In ground-based optical observations, accurately understanding effects of the scattering is a significant concern that has been
- 14 highlighted for decades. Despite this long-standing recognition, our comprehension remains largely qualitative. The
- 15 quantitative analysis presented in this study could offer valuable new insights, yet there are notable discrepancies between the
- 16 simulation and measured results. Additionally, the study indicates that verifying effects beyond scattering is not feasible. These
- 17 factors obscure the extent to which this research has enhanced our previous understanding at a qualitative level. I believe
- 18 novelty of the study lies in advancing quantitative understanding, so please elaborate on this aspect more specifically. While
- 19 the current text contains the desired content, a clearer expression of its novelty would improve the paper. For instance, revising
- 20 the following points could enhance the overall quality.
- 21 "With fixed initial speeds, the simulation reproduced the temporal characteristics of the atypical winds, demonstrating that
- 22 scattering may contribute to these intense horizontal differences and downwelling. The simulation also shows that the
- 23 scattering-induced biases have directional inhomogeneity with characteristics linked to the location and background line-of-
- 24 sight speed of the brighter airglow region. The accuracy of the simulation is limited by the accuracy of airglow observations
- 25 and atmospheric optical depth."
- 26 To emphasize the novelty in this field or to differentiate from our previous level of understanding, I believe it is crucial to
- 27 present results in a more quantitative manner rather than a qualitative one. In the abstract, I selected these sentences because
- 28 they contain potential information that highlights the novelty of this study. By revising them to include more qualitative details,
- 29 such as the possible percentage contribution of the scattering process, more specific information on directional inhomogeneity,
- 30 and additional data from other measurements that we can enhance the simulation quality in future work. Although the word

- 31 limit in the abstract makes it challenging to accommodate all my requests, please attempt to revise these points by
- 32 incorporating more quantitative information. In the text, it is beneficial to emphasize the novelty by adding these points, even
- 33 though some are already mentioned but dispersed throughout the text.
- 34 Key revisions:
- 35 (1) added the simulated horizontal-difference magnitudes:
- 36 L18 (Abstract) "With fixed initial speeds, the simulation reproduced the temporal characteristics of the atypical winds,
- 37 demonstrating that scattering may contribute to these intense horizontal differences and downwelling." to "With fixed initial
- 38 winds (100 m·s-1 westward, 400 m·s-1 southward, zero vertical wind), the simulation reproduces horizontal differences of
- 39 approximately 400 m·s-1 on May 10th and 100 m·s-1 on Oct 10th, both capturing the temporal characteristics of the atypical
- 40 winds.
- 41 Similar details have been added to L459 (Conclusions).
- 42 (2) added the relative changes in LOS speed among different directions:
- 43 L20 (Abstract) "The simulation also shows that the scattering-induced biases have directional inhomogeneity with
- 44 characteristics linked to the location and background line-of-sight speed of the brighter airglow region." to "The simulation
- 45 shows that scattering-induced biases on line-of-sight speed take their sign from the brighter region, while their magnitude
- 46 varies directionally with the angle to that region: at 45° elevation, biases 135–180° azimuth away exceed those in the brighter
- 47 region by more than 10 times."
- 48 L346 (Discussions 4.1) "The northward LOS speed changes slightly, the southward speed changes the most and nearly
- 49 reverses, while the eastward and westward speeds are intermediate." to "Across all azimuths, the changes of LOS speed share
- 50 the same sign, but their amplitude scales with the angle to the northward direction. Relative to the roughly 300 m·s-1 LOS
- 51 speed change in the southward direction, the eastward and westward directions each attain about 60 %, whereas the northward
- 52 variation remains below 10 %. The scattering model shows LOS speed changes in dimmer airglow regions are more than 10
- 53 times those in the brighter zone."
- 54 (3) added the relative changes in simulated scattering biases under different experimental conditions
- 55 **L22 (Abstract)** "The accuracy of the simulation is limited by the accuracy of airglow observations and atmospheric optical
- 56 depth" to "Limited by uncertainties in airglow images and optical depth of model inputs, the simulation incurs numerical errors
- 57 of roughly 75 % during some periods. Effective correction of the scattering impact will require improved accuracy of model
- 58 inputs in the future."

- L396 (Discussions 4.2) "When the optical depth is artificially increased to a higher value, such as around 0.6, the model more closely matches the Oct. observations." to "When the optical depth is artificially raised to 0.5, the model produces a meridional wind difference exceeding 400 m·s-1 at 18:30 UT, Oct. 10th, roughly triple the value obtained with an optical depth of 0.2 and in much closer agreement with the observations. We find that the model underestimates scattering effects when the optical depth is low. Once the optical depth reaches 0.6 or higher, the simulated wind bias accelerates nonlinearly until the model diverges."
- **L411 (Discussions 4.2) add** "In our experiments, without the stray-light correction in Appendix B, the airglow brightness gradient flattens slightly, the model becomes more inert, and the simulated horizontal wind differences shrink by about 30 %."